# Laboratory Measurements of the Performances of the Sweeping Langmuir Probe Instrument Aboard the PICASSO CubeSat

Sylvain Ranvier [1], Jean-Pierre Lebreton [2]

[1]Royal Belgian Institute for Space Aeronomy (BIRA-IASB), 1180 Brussels, Belgium
[2]LPC2E, CNRS/University Orleans/CNES, 45071 Orleans, France

*Correspondence to*: Sylvain Ranvier (sylvain.ranvier@aeronomie.be)

**Abstract.** The Sweeping Langmuir Probe (SLP) is one of the instruments on board the triple unit CubeSat PICASSO, an ESA in-orbit demonstrator launched in September 2020, which is flying at about 540 km altitude. SLP comprises 4 small cylindrical probes mounted at the tip of the solar panels. It aims at performing in-situ measurements of the plasma parameters (electron

density and temperature together with ion density) and of the spacecraft potential in the ionosphere. Before the launch, the instrument, accommodated on an electrically representative PICASSO mock-up, has been tested in a plasma chamber. It is shown that the traditional orbital-motion-limited collection theory used for cylindrical Langmuir probes cannot be applied directly for the interpretation of the measurements because of the limited dimensions of the probes with respect to the Debye length in the ionosphere. Nevertheless, this method can be adapted to take into account the short length of the probes. To

reduce the data downlink while keeping the most important information in the current-voltage characteristics, SLP includes an on-board adaptive sweeping capability. This functionality has been validated both in the plasma chamber and in space and it is demonstrated that, with a reduced number of data points the electron retardation and electron saturation regions can be well resolved. Finally, the effect of the contamination of the probe surface, which can be a serious issue in Langmuir probe data analysis, has been investigated. If not accounted for properly, this effect could lead to substantial errors in the estimation of

the electron temperature.

## 1 Introduction

The Sweeping Langmuir Probe (SLP) instrument, one of the two instruments on board the Pico-Satellite for Atmospheric and Space Science Observations (PICASSO), has been developed at the Royal Belgian Institute for Space Aeronomy. PICASSO, an ESA in-orbit demonstrator launched in September 2020, is a triple unit CubeSat of dimensions 340.5x100x100 mm.

The PICASSO mission ended in June 2022 because of issues with the platform. Although a limited amount of time has been allocated to the payload commissioning, it has been possible to validate all the measurement and diagnostic modes of SLP.

The expected plasma parameters along the orbit of PICASSO are given in Table 1 (Minow et al., 2004).

**Table 1.** Expected plasma parameters

| Plasma parameters | Minimum | Maximum |
| --- | --- | --- |

| | | |
|---|---|---|
| Plasma density (#/m³) | $5 \times 10^8$ | $10^{13}$ |
| Electron temperature (K) | 600 | 10 000 |
| Debye length (m) | $5.4 \times 10^{-4}$ | 0.31 |

Langmuir probe instruments have been used for decades on board large/medium-size satellites to measure ambient plasma properties (electron density and temperature together with ion density) (Boggess et al., 1959; Brace et al., 1965; Lebreton et al., 2006; Eriksson et al., 2007; Andersson et al., 2015; Knudsen et al., 2017) but their operation on board smaller platforms, such as nano-satellites, raises several issues in addition to miniaturisation and drastic reduction of power consumption. The limited conducting area of the spacecraft leads to spacecraft charging and drift of the instrument's electrical ground during the measurements, which can lead to unusable data (Ranvier et al., 2017, Leon et al. 2022). Furthermore, the limited telemetry bandwidth available on nano- to small satellites requires the use of specific measurement and data processing approaches. Finally, the Orbital-Motion-Limited (OML) collection theory (Mott-Smith and Langmuir, 1926) traditionally used for Langmuir probes cannot be applied directly because of the limited dimensions of the probes with respect to the Debye length in the ionosphere.

On the other hand, a nano-satellite (or electrically representative mock-up) embarking a Langmuir probe instrument can fit into plasma chambers, which is hardly feasible (if not impossible) for large satellites. This allows studying the performance and limitations of the instrument as well as the coupling between the Langmuir probe, the S/C and the surrounding plasma in laboratory.

## 2 SLP

SLP is a four-channel Langmuir probe instrument comprising four short cylindrical probes accommodated at the corners of the solar panels. Its measurement principle is based on the conventional Langmuir probe theory (Merlino, 2007). By sweeping the potential of one probe with respect to the spacecraft floating potential and measuring the current collected, the instrument acquires a current-voltage (I-V) characteristic from which the electron density and temperature, ion density and S/C (spacecraft) potential are retrieved. The analysis of the measurements are performed in three regions of the I-V curve: ion saturation, electron retardation and electron saturation regions. A typical I-V characteristic of such a probe is illustrated in Figure 1. The ion density, electron temperature and electron density are derived from the ion saturation, electron retardation and electron saturation regions, respectively.

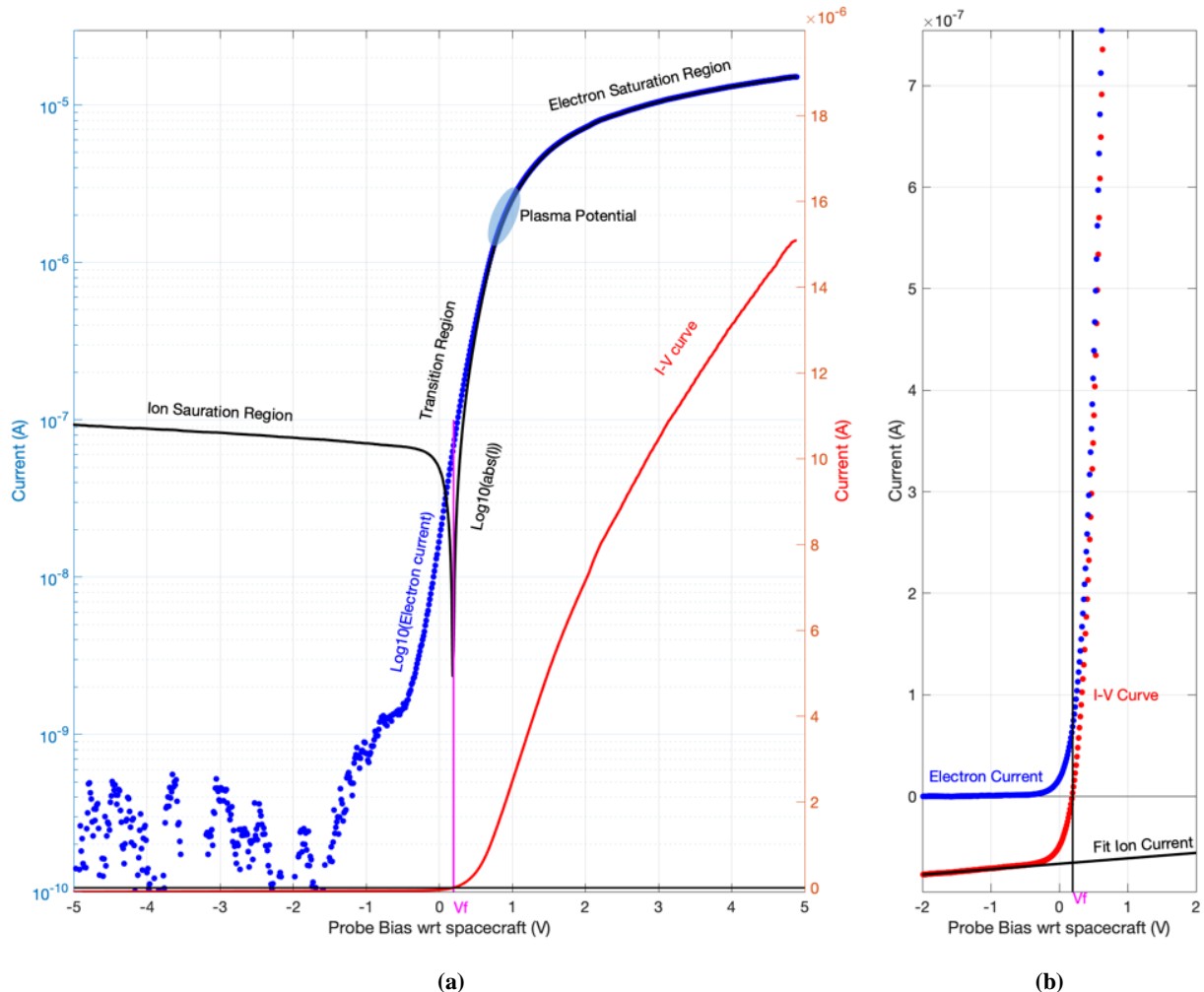

**(a)**                                                   **(b)**

**Figure 1. Langmuir probe current-voltage characteristic measured by SLP in the ionosphere on 30.01.2021 at 13:03:47 UTC. (a) full sweep with the measured signal in red (linear scale) and black (semi-log scale). The computed electron current is plotted in blue (semi-log scale). (b) zoom on -2 V to +2 V bias range (linear scale only) with the linear fit to the ion current plotted in black. The computed floating potential *Vf* is shown in magenta on both sub-figures.**

60

In addition, to overcome the S/C charging issue, while one probe is used as a traditional Langmuir probe as described above, a second probe (can be any of the three remaining probes) measures the floating potential with respect to the S/C potential. By combining the two measurements, consistent I-V characteristics can be retrieved (Ranvier et al., 2019, Leon et al. 2022). In nominal mode, SLP sweeps the potential of the probes from -5 V to +13 V with respect to the S/C potential in order to retrieve

65   the electron density and temperature, together with the S/C potential and the ion density. The sampling frequency is fixed at 10 kHz and the maximum sweeping frequency is about 50 sweeps/s.

The limited downlink bandwidth does not allow performing linear voltage sweeps with very fine steps in nominal mode. Instead, the three regions (ion saturation, electron retardation and electron saturation) are measured with different step sizes. Ion and electron saturation regions are measured with large voltage step size (in the order of 1 V or more). The electron retardation region is measured with smaller step size, which depends on the electron temperature: this region is measured with a fixed number of steps, but the span is adapted as a function of the temperature. On board, the inflection point of the I-V curve (i.e. the bias for which the first derivative of the curve is maximal) which separates the electron retardation and the electron saturation regions (the plasma potential) is determined by computing the first derivative of the curve and selecting the highest value (the maximum). This is performed after each sweep and used to compute the span of the fine step region of the next sweep, as follows: the first point of this region is set by a parameter of the command sent to the instrument, but the last point of this region is defined by adding a number (another parameter of the command) of data points after the computed inflection point, as depicted in Figure 2. The step size of the electron saturation region is then also adjusted automatically. The use of additional data points ensures the fine sampling of the full retardation region, even if the inflection point is computed with some inaccuracy (e.g. due to noise). There is no limitation on the minimum number of iterations needed to accurately determine the inflection point when the electron temperature decreases (inflection point shifting toward the retardation region). When the temperature is increasing (inflection point shifted toward the electron saturation region), the number of iterations needed to accurately define the inflection point is determined by the number of additional data points. Increasing the number of those additional data points will decrease the number of iterations needed to accurately determine the inflection point, but, at the same time, it will increase the data volume. Therefore there is a trade-off between data reduction efficiency and time response of the system. A running average (with window size a parameter of the command) is used to improve the robustness of the algorithm against the noise. Since the determination of the inflection point is done by computing the first derivative, the instrument current offsets (if any) will not impact its accuracy. Given that the algorithm defining the bias steps is fully deterministic, only one number (the on-board computed inflection point, which is sent together with the measured current) is needed to rebuild the sweep steps on ground and to compute the complete I-V curve. To get an order of magnitude, when the fine step region is sampled with 30 steps, the step size ranges from about 10 mV to 150 mV for electron temperatures of 600 K and 10.000 K, respectively. This method is different from the one used for the LPW instrument on MAVEN (Andersson et al., 2015) where a look-up table for the sweep steps (large step sizes in electron and ion saturation regions and small step sizes in the retardation region) is used and the sweep is centered on the location where the previous measured current changed sign plus an offset. The advantage of the approach used for the LPW as compared to the one used for SLP is that the beginning of the fine step region is adjusted automatically, whereas it is fixed for SLP. On the other hand, the approach used for SLP allows sampling the retardation region with a given number of points, leading to a given relative resolution, which is valid for a very broad range of electron temperature, i.e. for retardation regions with very different width.

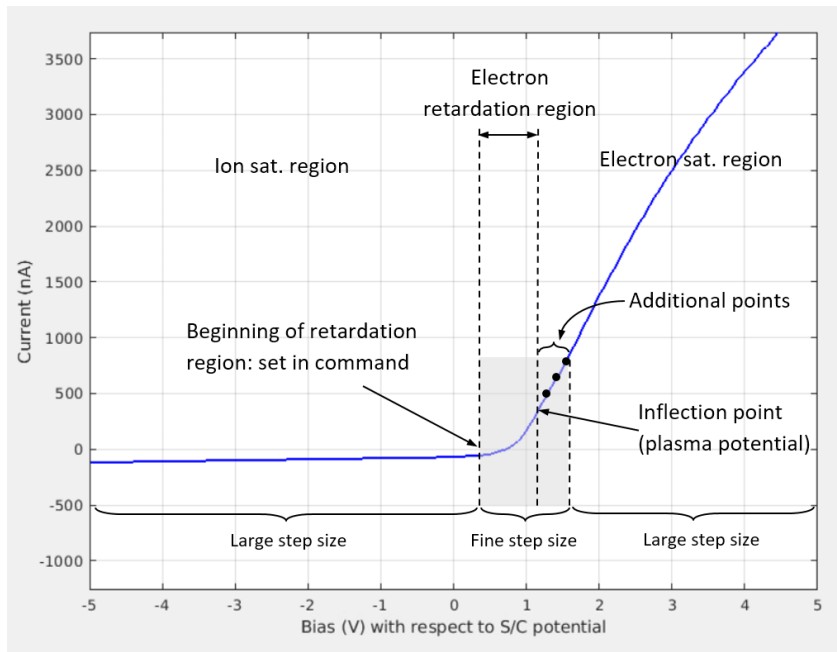


**Figure 2. Example of sweep using the adaptive algorithm.**

In another mode, the instrument measures only in the electron saturation region at a higher rate, measuring electron density

with better temporal, hence spatial resolution in order to resolve fine plasma structures like those presented in (Hoegy et al., 1982). This later operating mode is based on the principle described in (Bekkeng et al., 2010). Although the telemetry is limited, the raw data are downloaded to the ground because the measured current-voltage characteristics contain more information than only four parameters (electron density and temperature, ion density and S/C potential). The four probes of SLP are mounted on the corner of the deployable solar panels, which act as deployable booms, as depicted in Figure 3. This

configuration ensures that at least one probe is out of the S/C wake at any time, in addition to providing redundancy.

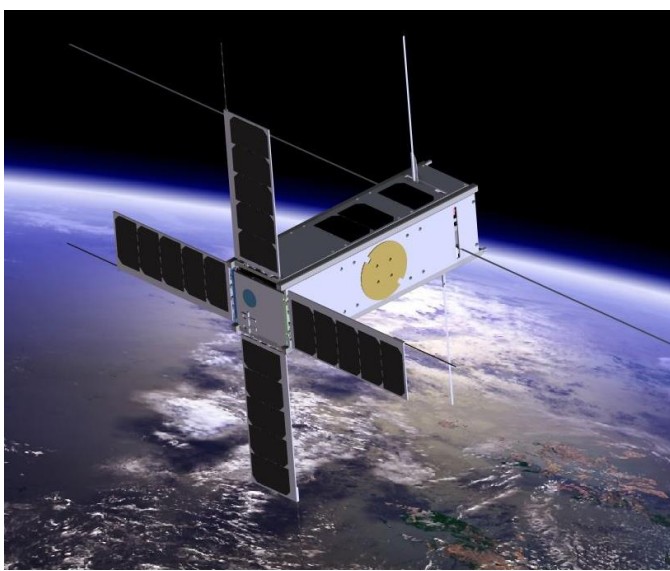

**Figure 3. SLP probes on PICASSO (credit ESA).**

The probes, coated with gold, are 40 mm long Ti tubes of 2 mm diameter. They are attached to the extremity of the solar panels
via a 40 mm long boom, as depicted in Figure 4.

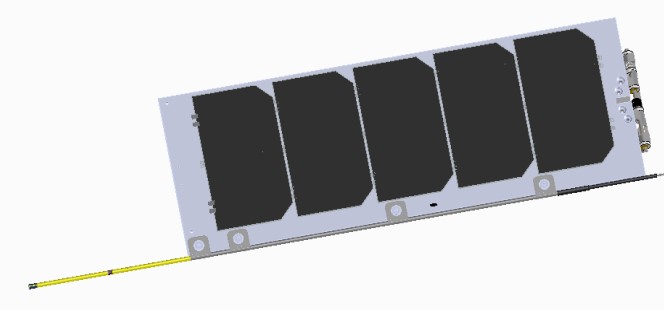

**Figure 4. 40 mm long probe sticking out of a 40 mm long boom (both in yellow) at the extremity of the solar panel. Both the probe and the boom are 2 mm outer diameter gold coated Ti cylinders.**

On the solar panels, the five solar cells are covered with traditional coverglass. The interconnects solder joints between the cells are covered with a dielectric coating but the interconnects themselves and the edges of the metalized layer below the cells are not covered. This leads, for each cell, to 7.1 mm² of exposed conductive surface that is at another potential than the S/C chassis. To minimize the effects of the solar cells on the plasma surrounding the probes, the five solar cells are mounted on the solar panels in such a way that their potential is increasing as the cells are closer to the S/C body, as depicted in Figure 5.
Therefore, the closest non-grounded conductive surface, which is at about + 3 V with respect to the S/C GND, is 100 mm away from the probes. It should be noted that between these non-grounded conductive areas and the probes there are about 2000

mm² of grounded surface (at the tip and on the side of the solar panels) which acts as a shield for the probes together with the 40 mm-long boon which is also grounded.

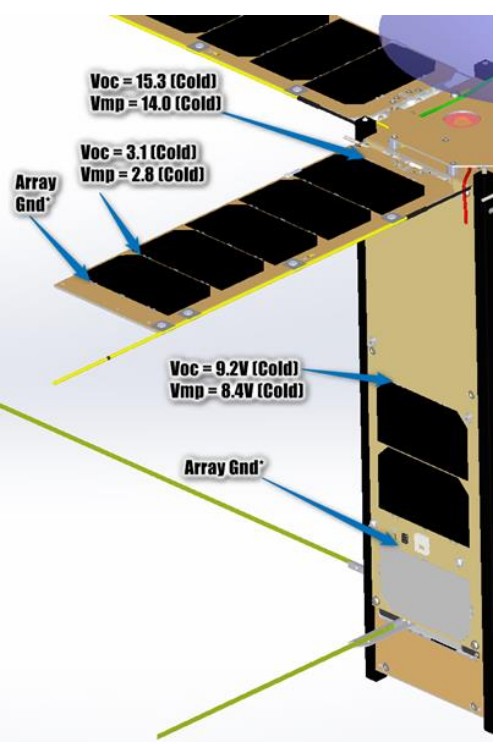


**Figure 5. Location of the solar cells and interconnects on the solar panels (credit AAC Clyde Space AB).**

Due to constraints related to the small diameter of the probe and coax cable as well as the mechanical complexity inside the probe and boom, it was not possible to include a guard electrode on the probe. Furthermore, a guard electrode would increase

the collecting area which would make the spacecraft charging effect more severe.

There will be no active on-orbit cleaning of the probe surface. First, since PICASSO flies on a high inclination orbit, it is expected that natural ion and electron sputtering will occur at high latitude where the spacecraft will encounter energetic ions and electrons. This natural sputtering seemed to be sufficient for several Langmuir probe instruments which did not show signs of contamination (Tiros-7, Explorers 17, 22, 23, 31, 32, Alouette-II, ISIS-1 and ISIS-2) (Brace, 1998). Furthermore, for a

significant part of the orbit, the temperature of the probe may reach about 200 °C, which should help cleaning (Amatucci et al., 2001). Finally, an active on-orbit cleaning would have significantly increased the complexity of the instrument and created additional constraints.

## 3. Functional test in laboratory plasma chamber

Because the integration of PICASSO was not completed before the test, the functional test was performed using an electrically

representative satellite mock-up. Since PICASSO displays a conducting surface of at least 200 cm$^2$ normal to the velocity vector at any moment, the S/C model is a cube of aluminium, each side 141.4 mm long. The four probes are laying in the horizontal plane, as depicted in Figure 6. Two probes, coated with gold, are 40 mm long and stick out of a 60 mm long boom while the two other probes, coated with TiN, are 50 mm long and stick out of a 50 mm long boom. All probes and booms are 2 mm diameter Ti tubes. The S/C mock-up is coated with conducting carbon-based paint.

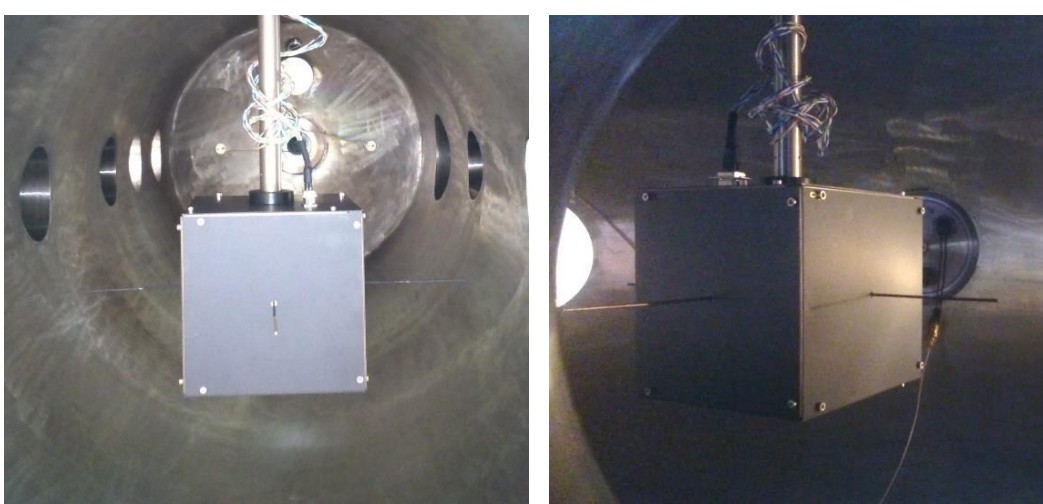


**Figure 6. Picture of the electrically representative PICASSO mock-up in the plasma chamber.**

The test was performed in the plasma chamber at ESA/ESTEC, Noordwijk, NL. The chamber is a 250 cm-long cylinder of 80 cm inner diameter. There is an external motor that allows rotating the test model inside the chamber.

To measure accurately the I-V characteristics in a wide range of currents, SLP electronics is very sensitive and covers a wide dynamic range. During a sweep (acquisition of an I-V curve) the acquired current can be very faint (a few hundreds of pA or less) and the measured floating potential lies between a few mV and a few V. Therefore, special care must be taken when performing functional tests because the noise originated from, or picked up by the electrical ground support equipment (EGSE) can severely affect the measurement data and make it nearly unusable. Filtering or averaging the data to reduce the noise must

be done only scarcely since it would decrease the time accuracy and hide numerous transient effects that are of foremost interest. Therefore, based on the experience acquired during previous tests in a similar environment, a specific EGSE has been designed, with special care given to the isolation of the control PC and the grounding implementation, as can be seen in Figure 7.

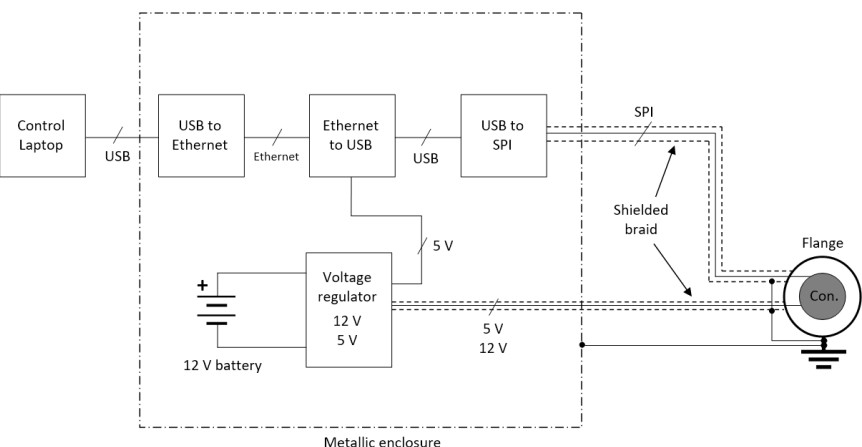

**Figure 7. EGSE to test SLP and the PICASSO mock-up in the plasma chamber.**

## 4. Application of the OML theory

The OML collection theory can be applied if, *inter alia*, the cylindrical probe radius is small compared to the Debye length and the length of the probe is much longer than the Debye length (in practice more than 10 times the Debye length). If all assumptions for the OML theory are met, the equations (1) and (2) below can be used to determine the electron density and
temperature from the I-V characteristics. If those assumptions are not met, then the equations must be adapted. In principle, the electron retardation region obeys the OML theory, irrespective of the probe geometry, hence the electron temperature can always be retrieved, if one can properly isolate the electron current contribution to the I-V curve.

Given the limited length of the probes of SLP or of other instruments flying on nano-satellites such as the one reported in (Bekkeng et al., 2010), the above mentioned requirements cannot be fulfilled. Therefore, the OML theory cannot be applied
directly for such instruments. The end effect at the tip of the probe leads to an ellipsoidal sheath, in-between a cylindrical and a spherical one. This phenomenon has been simulated with SPIS (Spacecraft Plasma Interaction System), a particle-in-cell (PIC) simulator software (Thiebault et al., 2015). The simulation model is a cube of aluminium (AL2K - Aluminium according to NASCAP-2k), each side 141.4 mm long, similarly to the model that has been tested in the plasma chamber, with a 40 mm long probe in aluminium (AL2K) located 60 mm away from the cube. The radius of the probe is 2 mm. The plasma density is
set to 2e10 /m³ and the electron and ion temperature to 0.05 eV. The simulation is performed in flowing plasma (V = 7.5 km/s, perpendicular to the probe), without taking into account photoemission (worst-case in terms of S/C charging due to Langmuir probe). The absolute spacecraft capacitance is set to 200 pF, ions are treated as PIC and electrons are treated as a Maxwell Boltzmann equilibrium distribution.

As it can be seen from the plasma potential maps displayed in Figure 8, there is a clear end effect at the tip of the probe. When
the applied bias is 2 V, the sheath around the probe is ellipsoidal, as indicated by the 0.05 V iso level curve. It should be noted that, as the bias increases, the influence of the S/C potential (which is more and more negative with respect to the plasma

potential) becomes more and more visible on the shape of the sheath around the probe. In figures 8(a), 8(b) and 8(c), 2 V, 4.75 V and 10.25 V, respectively, are applied to the probe (with respect to the S/C chassis), which, due to S/C charging effect, represent 1.25 V, 1.91 V and 2.94 V with respect to the plasma potential, respectively. The fact that the main effect is the influence of the spacecraft potential rather than the wake due to the probe itself as reported in (Ergun et al., 2021) comes from the fact that the probe to S/C current collection area ratio is much larger for SLP on PICASSO than for LPW on MAVEN, which leads to more severe S/C charging effects for SLP. In addition, given the short distance between the S/C body and the sensor (a few Debye lengths) and the fact that there is no guard, the S/C potential effect dominates the wake effect on the I-V characteristics for SLP.

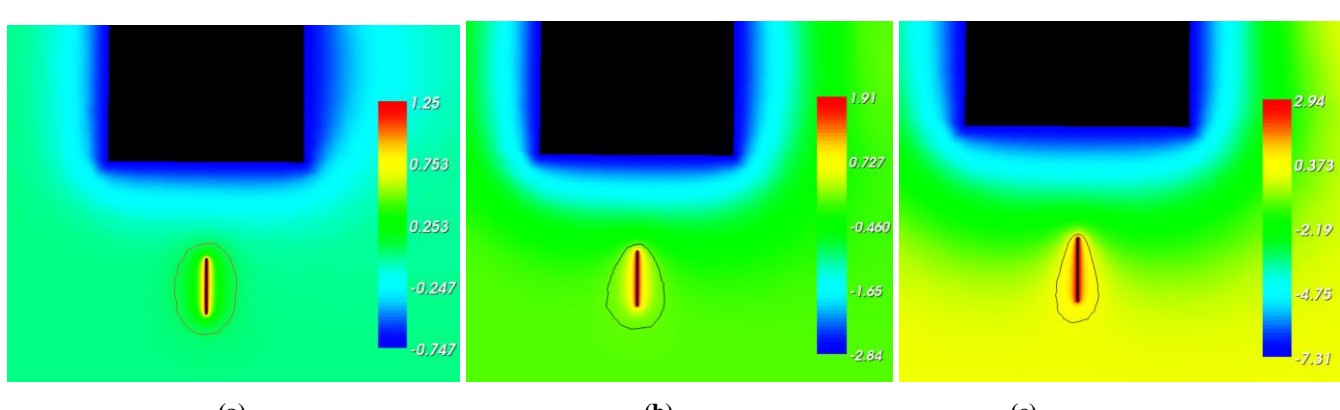

|          (a)          |          (b)          |          (c)          |

**Figure 8. Plasma potential around the S/C mock-up when a bias of 2 V (a), 4.75 V (b) and 10.25 V (c) is applied on the probe, together with the iso level curve at 0.05V (linear scale, in V).**

An analytical general expression for the current in the electron saturation region (Chen, 1965) can be written as Eq. (1):

$$I_e(\phi) = I_{the}\left(1 + \frac{q_e(\phi - \phi_p)}{k_B T_e}\right)^\gamma \tag{1}$$

where $\phi$ is the applied bias, $\phi_p$ is the plasma potential, $q_e$ is the charge of an electron, $k_B$ is the Boltzmann constant, $T_e$ is the electron temperature and $\gamma$ = 0, 0.5 and 1 for planar, cylindrical and spherical probes, respectively. $I_{the}$, the random thermal electron current to a probe in a Maxwellian plasma, is given by Eq. (2):

$$I_{the} = n_e q_e A \sqrt{\frac{k_b T_e}{2\pi m_e}} \tag{2}$$

where $ne$ is the electron density and $A$ is the surface area of the probe.

For short cylindrical probes, because of the end effect and the ellipsoidal shape of the sheath, the $\gamma$ exponent is not 0.5 as for infinitely long probes, but lies between 0.5 and 1, somewhere in between an infinitely long cylindrical probe and a spherical probe.

The deviation from the infinitely long probe case depends on the length of the probe with respect to the dimensions of the sheath around it, and thus the plasma parameters (density and temperature). Consequently, it is of utmost importance to compute the γ exponent as described in (Lebreton et al., 2006) for each I-V characteristic in order to accurately retrieve the electron density.

As it can be seen on the measured I-V characteristic plotted in Figure 9 in blue (discrete dots), where the instrument was grounded to isolate the end effect from S/C charging effects, the γ exponent is not 0.5 (square root shape) but has been found to be 0.69. Using this value, the plasma density was computed. The plasma parameters for this measurement were $n_e = 4.2e10/m^3$ and $T_e = 650$ K. For comparison, the theoretical I-V characteristics of the electron saturation region from an infinitely long cylindrical probe (γ = 0.5), a spherical probe (γ = 1) and the γ = 0.69 fit are plotted in Figure 9 in red, black and

magenta, respectively. The magenta curve (γ = 0.69 fit) satisfactorily overlap the blue curve (measurement data points). The results presented in this paper qualitatively agree with those reported in (Ergun et al., 2021) and (Marholm et al., 2020), where it is shown that in the ionosphere, for cylindrical probes with finite length, γ lies between the one of a perfect cylinder (0.5) and the one of a perfect sphere (1).

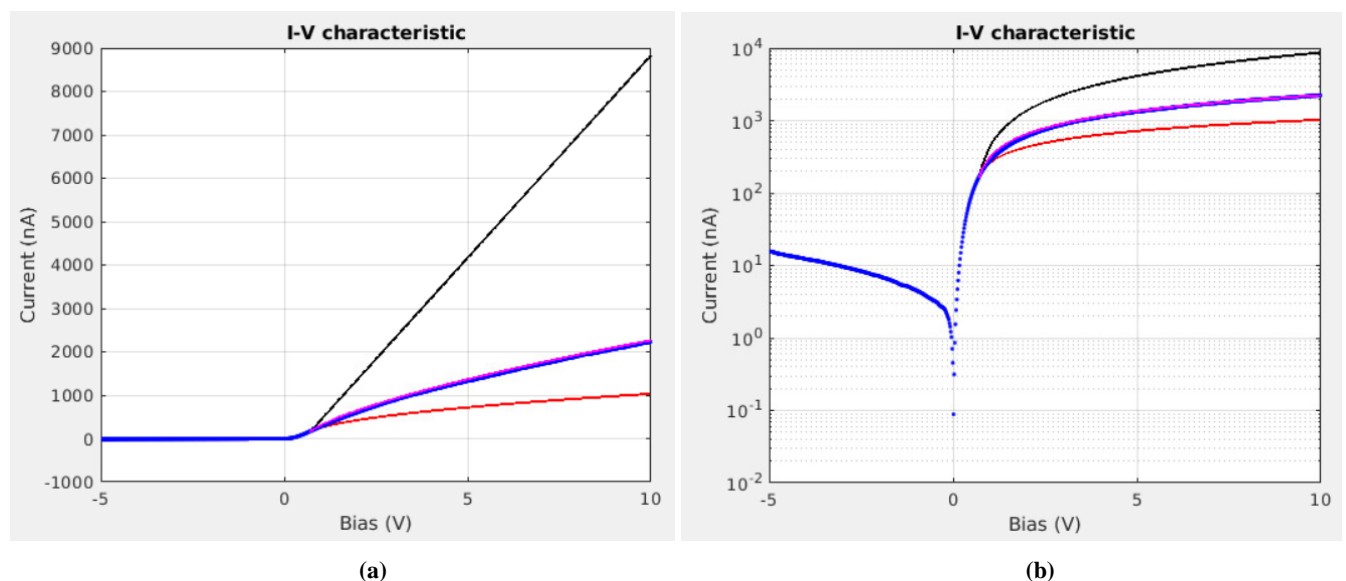


(a)                                                                                  (b)

**Figure 9. I-V characteristic with 840 data points. The blue dots are the measured data. The red, black and magenta lines are the computed data with γ= 0.5, γ= 1 and γ= 0.69, respectively. (a) linear scale, (b) semi-logarithmic scale.**

## 5. Validation of the adaptive sweeping method

As discussed in Section 2, to reduce the amount of data to be downlinked while keeping the most relevant information, the sweeps can be performed using adaptive non-equidistant steps, which is called the nominal mode. From the I-V characteristic,

measured in the laboratory plasma chamber, plotted in Figure 10, it can be seen that with only 43 data points sampled at the proper biases the I-V characteristic is well resolved in the electron retardation and the electron saturation regions. The retrieved electron density and temperature are comparable to the ones retrieved from the 1000-data-point sweeps. In this figure, derived

from an upward sweep (dV/dt positive), it can be seen that there is a non-monotonic behaviour between the ion saturation and the electron retardation regions. This effect is due to the capacitance of the front end and the delay between samples used during the test in the plasma chamber. This phenomenon is better visualised in Figure 11, which is a plot of the I-V characteristic when a 50 Mohm resistor is connected to the probe instead of the plasma. The red data points are the measured currents when the delay between samples is set to 300 µs, as for the measurement displayed in Figure 10. For the blue data

points, the delay between samples has been increased to 600 µs. The black curve is the expected ideal curve. The standard deviation of the difference between the data points and the ideal curve are 6.7 % and 1.1 % for 300 µs and 600 µs delay between samples, respectively. Therefore, by increasing the delay between samples (which is one of the measurement parameters for SLP), the effect of the capacitance can be decreased to an acceptable level.

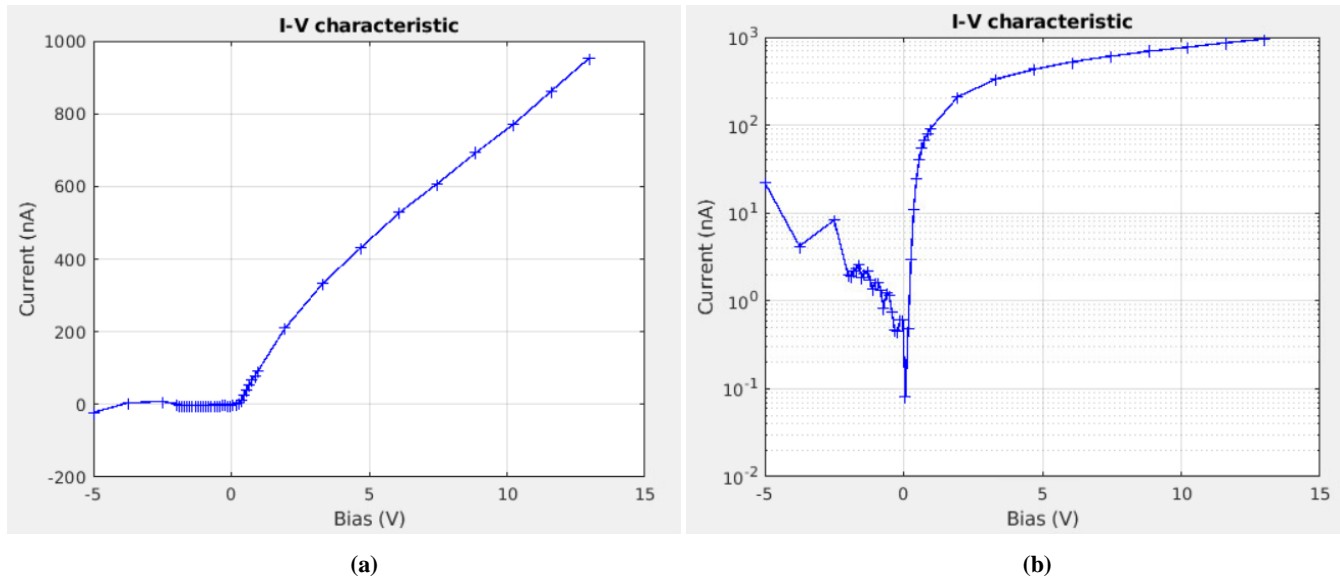

(a)                                          (b)


**Figure 10. I-V characteristic with 43 data points measured in the laboratory plasma chamber. (a) linear scale, (b) semi-logarithmic scale.**


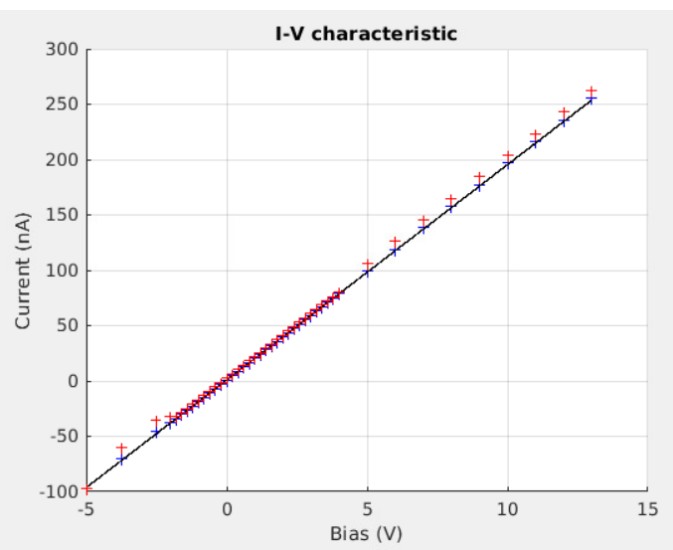

**Figure 11. Measured I-V characteristics when the probe is connected directly to a 50 Mohm resistor. Delay between samples set to 300 µs (red) and 600 µs (blue). For reference, the expected ideal curve is plotted in black.**

Figure 12 shows the inflection point calculated in real time by SLP after each sweep for a series of 19 consecutive nominal sweeps (one sweep per second). For the first sweep, the initial value is set by the operator when sending the command (2.5 V in this example). The number of additional data points in the fine step region is set to 3. It can be seen that the convergence is fast. The actual inflection point is reached at the third sweep, although the initial condition sent to the instrument (2.5 V) is relatively far from the actual value (0.46 V). It is worth noting that, for a stable laboratory plasma, once the inflection point is

correctly estimated, it remains constant for the rest of the series within 1%.

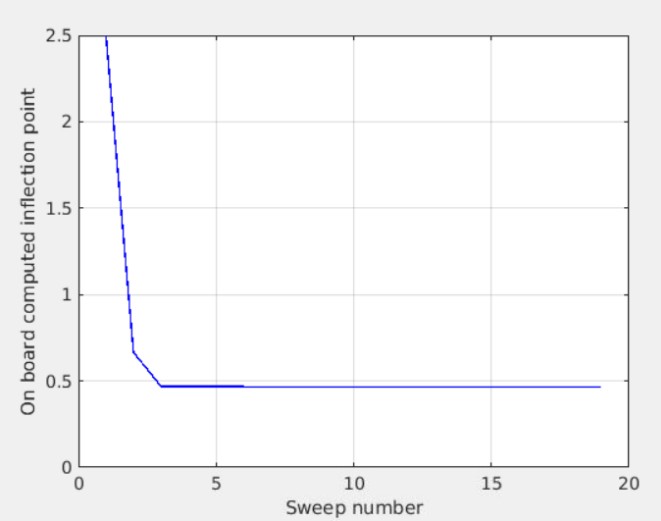

**Figure 12. Inflection point computed on board SLP after each sweep for a series of 19 consecutive nominal sweeps.**

A similar test has been performed during the in-flight commissioning of SLP for a series of 14 consecutive sweeps, 60 bias steps each, with a given starting inflection point of 1 V and the number of additional data points in the fine step region set to 5. The acquired I-V characteristic is plotted in Figure 13. To mitigate the effect of the internal capacitance visible in the laboratory test (see Fig. 10), a delay of 1.2 ms has been added between samples, leading to a total sweep duration of 78 ms. From Figure 14 it can be seen that, although the convergence is not as fast as for the laboratory test, it remains within 0.35 V

of the assumed target value (1.58 V) from the third sweep on.

The distortion of the I-V characteristics due to the adaptive sweep method depends on several parameters including the number of data points, the difference of the step sizes of the three regions and the delay between the data points. The effect of the latter parameter is clearly visible by comparing Figure 10 and Figure 13, which are sweeps performed with 300 µs and 1.2 ms delay between steps, respectively. Unfortunately, not enough measurements have been performed in the laboratory plasma chamber

to statistically quantify the change in the uncertainty of electron density and temperature as a function of those parameters. This assessment will be performed in a follow-up study.

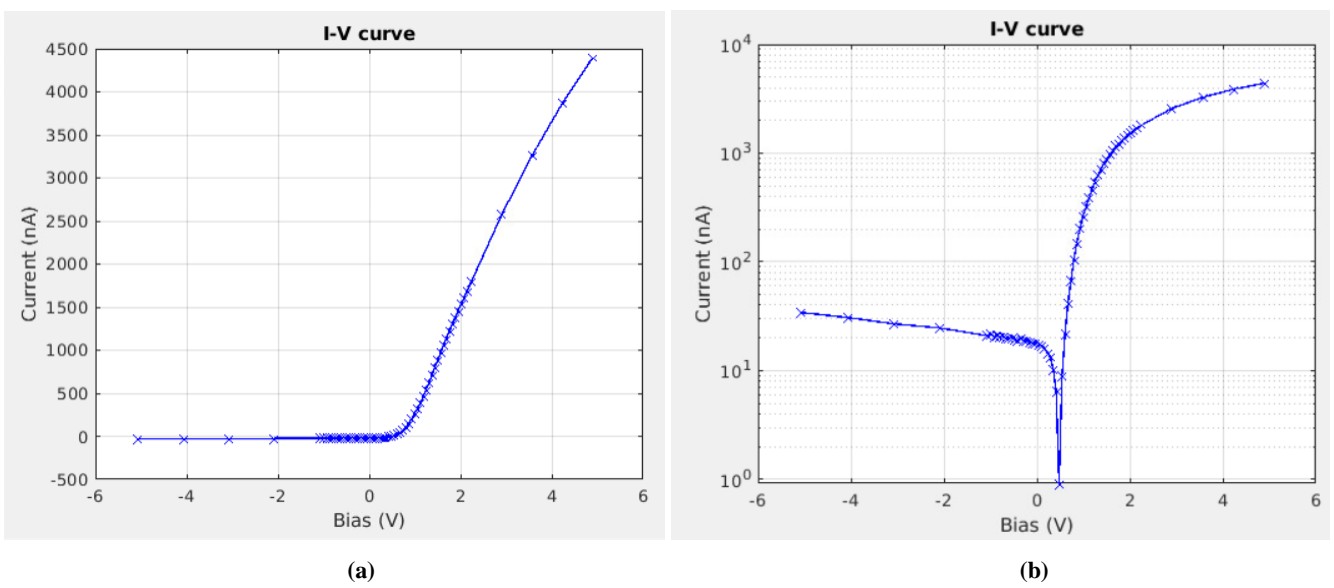

(a)                                                                      (b)

**Figure 13. In-flight measured I-V characteristic with 60 data points acquired on 30.01.2021 at 13:03:59 UTC. (a) linear scale, (b)**
**semi-logarithmic scale.**

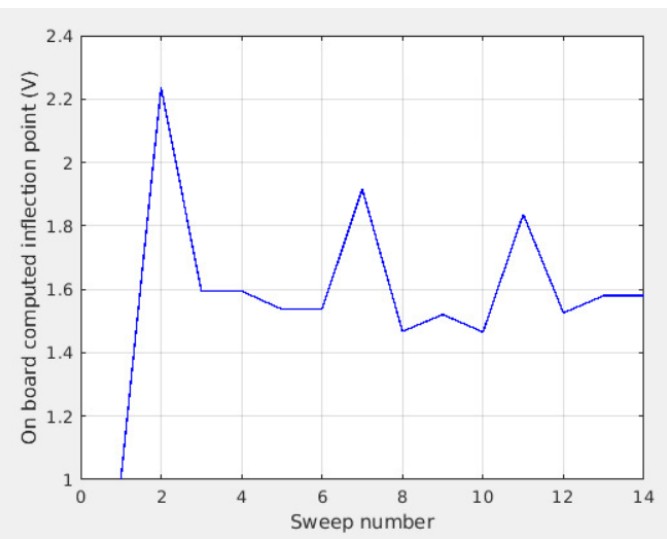

**Figure 14. Inflection point computed on board SLP after each sweep for a series of 14 consecutive sweeps with 60 bias steps. Data acquired in flight on 30.01.2021 from 13:03:53 to 13:04:06.**

## 6. Effects of the contamination

Surface contamination is a serious issue for Langmuir probe operation. When a probe is covered by a thin contamination layer, this layer act as a capacitor C with a resistor R in parallel that will charge and discharge with an RC time constant during or between consecutive sweeps (Fang et al., 2018). The same phenomenon can happen at the surface of the S/C body. This will decrease or increase the actual potential in contact with the plasma. For a typical sweep (e.g. from – 5 V to +13 V), the probe will collect mostly electron current because 1) there is a longer period of time in the electron saturation region than in the ion saturation region and 2) the electron current is at least 50 times larger than the ion current. Therefore, the contamination layer will charge negatively. This charging causes an hysteresis in the I-V characteristics of two-way sweeps (e.g. from -5 V to +13 V to -5 V in one sequence) which depends on the sweep duration. In addition, since during operation the sweeps are performed in series with only a few milliseconds between two consecutive sweeps, the contamination layer will charge not only during one sweep, but it will also charge over the series of sweeps, with a cumulative effect, until the "capacitor" is fully charged. This charging effect is shown in Figure 15 where I-V characteristics of a series of three consecutive two-way sweeps (from - 5 V to +13 V to -5 V in one sequence) of 1280 ms (640 ms increasing bias and 640 ms decreasing bias) duration with a delay of about 1 ms between two consecutive sweeps are plotted. It can be seen that 1) the contamination layer charges to - 0.9 V in three sweeps, 2) at a given probe potential the current is reduced and 3) because the curve is "shifted" towards the electron saturation region, the derived plasma potential becomes more positive.

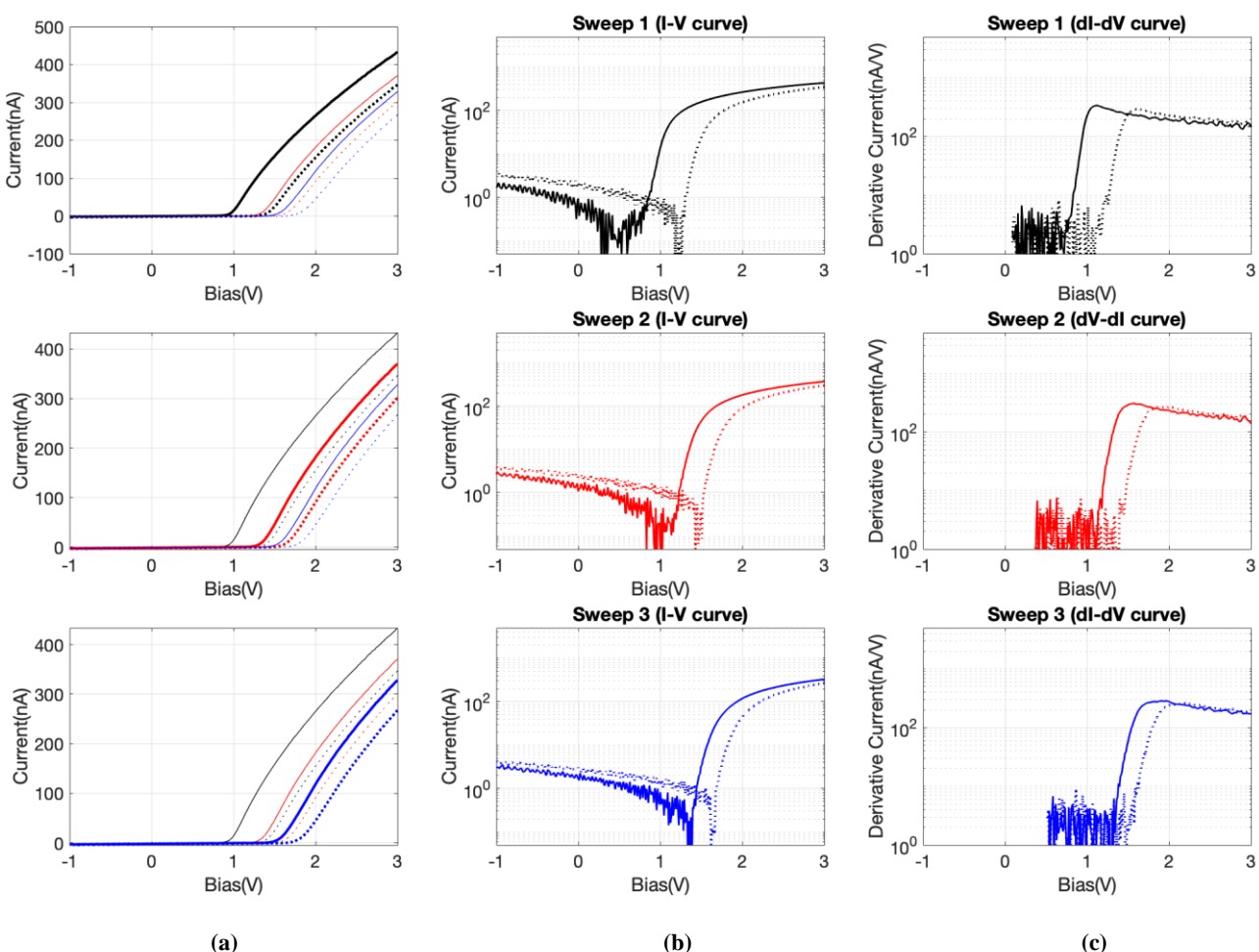

**Fig. 15. I-V characteristics of three consecutive two-way sweeps measured in the laboratory plasma chamber when the S/C GND is connected to the chamber chassis (zoom on -1 V to + 3 V bias range). The increasing and decreasing parts of the sweeps are in solid and dashed lines, respectively. The first, second and third sweeps are in blue, red and black, respectively. (a) the three consecutive two-way sweeps with the first, second and third sweeps in bold in the top, middle and bottom sub-figures, respectively (linear scale). (b) first, second and third sweeps in the top, middle and bottom sub-figures, respectively (semi-log scale). (c) first derivative of the current for the first, second and third sweeps in the top, middle and bottom sub-figures, respectively (semi-log scale).**

The fact that the charging of the contamination layer could be misinterpreted as an increase of electron temperature is illustrated in Figure 16, where the electron temperature, computed without taking into account the probe surface charging, is plotted as a function of the sweep number for a series of 9 consecutive linear sweeps from − 5 V to + 13 V.

Similar effects (i.e. reduced current at a given probe potential, the derived plasma potential becomes more positive, the derived electron density decreases and the derived electron temperature becomes hotter) have been reported in (Samaniego, J. I. et al., 2018) for several probe coatings (including Gold and DAG) both when the probes were exposed to oxygen-rich environments and without exposure to oxygen-rich environment. In this latter case, the sources of contamination were assumed to be the deposition of vacuum pump oil and moisture from the air, similarly to the contamination of the SLP probes reported in this

Section.

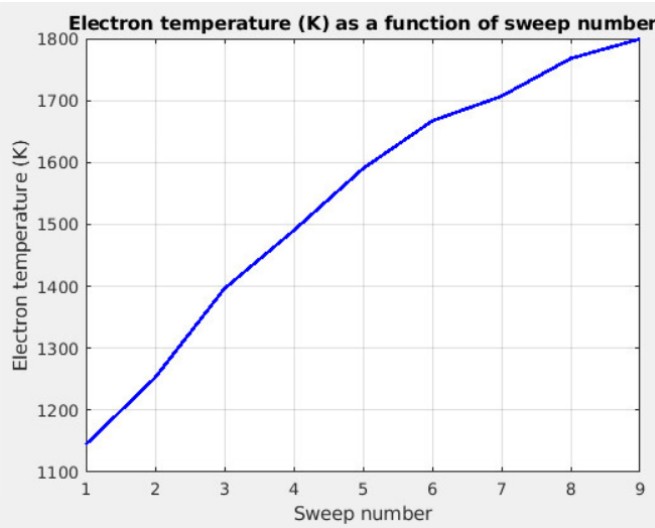

**Fig. 16. Electron temperature computed without taking into account the probe surface charging as a function of the sweep number**

**for a series of 9 consecutive linear sweeps from – 5 V to + 13 V.**

The time constant of this charging effect due to the surface contamination can be determined by plotting the current response to a voltage step for a sufficiently long time. Figure 17 displays the current measured by a probe when a bias step from 0 V to 13 V is applied and the S/C GND is connected to the chamber chassis, to avoid S/C charging. The drop of current from 1740

nA to 1330 nA corresponds to a voltage drop of 4 V across the contamination layer. This indicates that during a series of sweeps it will require several tens of seconds for the contamination layer to charge to its full value. It also implies that the first sweep, when the contamination layer is completely discharged, gives the most accurate I-V characteristic. For a space instrument, one possibility would be to periodically stop sweeping, wait for a given time (to let the probe-spacecraft system discharge), restart the sweep sequence and use the first sweep as a reference. If the plasma environment can be assumed to be

stable for a few tens of seconds, then the effects of contamination on the I-V curves can be accounted for in the calibration.

Measuring with multiple probes would allow acquiring continuous measurements data even if sweeps are stopped and restarted regularly (if this is done alternatively for both - or more – probes).

Another possibility is to perform increasing and decreasing sweeps with different duration as in (Lebreton, et al., 2006) and to
model the contamination layer as an R-C parallel circuit as in (Piel et al., 2001 and Fang et al., 2018). If properly modeled and combined with a database of PIC simulations of contaminated probe surfaces for different R, C and plasma parameters values, this could be used to retrieve the uncontaminated I-V curve. The study of the effect of the contamination, on a single sweep and on consecutive sweeps, is an on-going work that will be subject to a follow-up publication. Hence, in this article, we acknowledge that there are uncertainties in the determination of the plasma parameters, but their quantification is out of scope
of this paper.

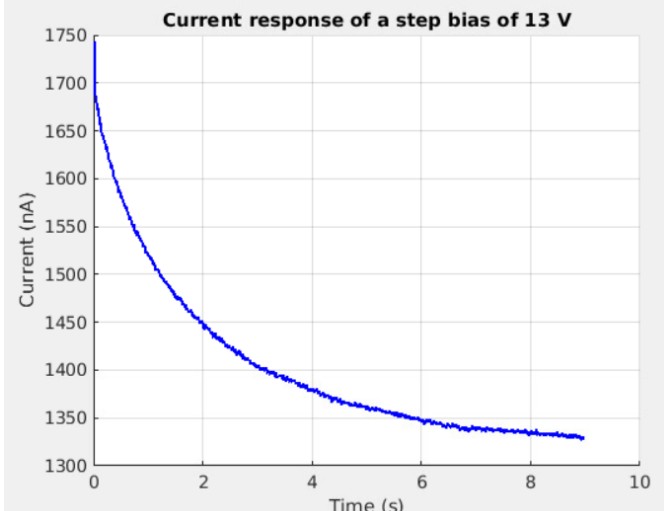

**Fig. 17. Current response to a 13 V bias.**

When operated in nominal mode, SLP is less subject to the charging of the probe surface because it spends less time in the
electron saturation region than during a linear sweep with equidistant bias steps. In order to reduce further this effect, the span of the electron saturation region can be reduced, e.g. limiting the sweep to + 5 V instead of + 13 V. In flight, consecutive two-way sweeps with different sweep durations have been performed to characterize the surface contamination, with similar results.

## 7. Conclusions

The SLP instrument, which flies on board the PICASSO nanosatellite, is a Langmuir probe instrument with four thin cylindrical
probes. Using a specific EGSE, an electrically representative mock-up of PICASSO has been tested in a laboratory plasma chamber. It has been shown that the traditional OML collection theory used for Langmuir probes cannot be applied directly

because of the limited dimensions of the probes with respect to the Debye length in the ionosphere. Nevertheless, the use of this theory can be adapted to take into account the short length of the probes. In addition, the onboard adaptive sweeping algorithm, which computes the inflection point of an I-V curve to define the bias steps of the next sweep has been validated both in the laboratory and in space. It is shown that with 60 data points the I-V curve can be very well resolved. Finally, the effect of the contamination of the probe surface, which can be a serious issue in Langmuir probe data analysis, has been investigated. Probe surface contamination can cause a drift of the I-V curve, resulting in substantial errors in the estimation of the electron temperature. With the limited data set acquired over 2 years, we will attempt to further study the effects of the contamination layer on the I-V curves to quantitatively take those effects into account for the determination of the plasma parameters, which would improve the accuracy of the Langmuir probe measurements.

### Author contributions

SR and J-PL jointly planned and carried out the measurement campaign. SR was responsible for operating SLP and providing the EGSE. J-PL operated the plasma chamber. SR performed the SPIS simulations and processed the measurement data. Both J-PL and SR actively contributed to the analysis and interpretation of the data. SR drafted the manuscript and J-PL reviewed it.

### Competing interests

The authors declare that they have no conflict of interest.

### Acknowledgements

The authors warmly thank Brian Shortt for making possible the tests in the plasma chamber at the ESA/ESTEC premises, as well as Thierry Beaufort and Ivo Visser for their very much-appreciated support during the test campaign. The authors would like also to thank the two referees for their very useful comments, which allowed them to improve the article.

### Financial support

This research was funded in part by the Solar-Terrestrial Center of Excellence, in part by the European Space Agency (ESA) under Grant 4000112430/14/NL/MH, and in part by the Belgian Scientific Policy Office (BELSPO).

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
