# Peer review of "Laboratory Measurements of the Performances of the Sweeping Langmuir Probe Instrument Aboard the PICASSO CubeSat"

_Geoscientific Instrumentation, Methods and Data Systems, 2022_

## Author Response (AR1)

**Referee 1**

Comment 1:
We added the following text:
"The distortion of the I-V characteristics due to the adaptive sweep method depends on several parameters including the number of data points, the difference of the step sizes of the three regions and the delay between the data points. The effect of the latter parameter is clearly visible by comparing Figure 9 and Figure 12, which are sweeps performed with 300 μs and 1.2 ms delay between steps, respectively. Unfortunately, not enough measurements have been performed in the laboratory plasma chamber to statistically quantify the change in the uncertainty of electron density and temperature as a function of those parameters. This assessment will be performed in a follow-up study."

Comment 2:
We added the following text:
"On board, the inflection point of the I-V curve (i.e. the bias for which the first derivative of the curve is maximal) which separates the electron retardation and the electron saturation regions (the plasma potential) is determined by computing the first derivative of the curve and selecting the highest value (the maximum). This is performed after each sweep and used to compute the span of the fine step region of the next sweep, as follows: the first point of this region is set by a parameter of the command sent to the instrument, but the last point of this region is defined by adding a number (another parameter of the command) of data points after the computed inflection point, as depicted in Figure 2. The step size of the electron saturation region is then also adjusted automatically. The use of additional data points ensures the fine sampling of the full retardation region, even if the inflection point is computed with some inaccuracy (e.g. due to noise). There is no limitation on how fast the inflection point can be determined accurately when the electron temperature decreases (inflection point shifting toward the retardation region). When the temperature is increasing (inflection point shifted toward the electron saturation region), the number of iterations needed to accurately define the inflection point is determined by the number of additional data points. Increasing the number of those additional data points will decrease the number of iterations needed to accurately determine the inflection point, but, at the same time, it will increase the data volume. Therefore there is a trade-off between data reduction efficiency and time response of the system. A running average (with window size a parameter of the command) is used to improve the robustness of the algorithm against the noise. Since the determination of the inflection point is done by computing the first derivative, the instrument current offsets (if any) will not impact its accuracy. Given that the algorithm defining the bias steps is fully deterministic, only one number (the on-board computed inflection point, which is sent together with the measured current) is needed to rebuild the sweep steps on ground and to compute the complete I-V curve. To get an order of magnitude, when the fine step region is sampled with 30 steps, the step size ranges from about 10 mV to 150 mV for electron temperatures of 600 K and 10.000 K, respectively. This method is different from the one used for the LPW instrument on MAVEN (Andersson et al., 2015) where a look-up table for the sweep steps (large step sizes in electron and ion saturation regions and small step sizes in the

retardation region) is used and the sweep is centered on the location where the previous measured current changed sign plus an offset. The advantage of the approach used for the LPW as compared to the one used for SLP is that the beginning of the fine step region is adjusted automatically, whereas it is fixed for SLP. On the other hand, the approach used for SLP allows sampling the retardation region with a given number of points, leading to a given relative resolution, which is valid for a very broad range of electron temperature, i.e. for retardation regions with very different width."

The number of additional data points used during the measurements presented in Fig. 10 and Fig. 13 has been added in the respective sections.
As suggested, we added the following reference: Andersson, L., et al., The Langmuir Probe and Waves (LPW) Instrument for MAVEN, Space Sci. Rev. (2015), 195:173–198, DOI 10.1007/s11214-015-0194-3

Comment 3:
Although the paper describes very well the non-linear effects linked to the fast sweeping of Langmuir probes, those effects are studied for sweep duration from 1.5 µs up to 0.2 ms, which is at least 100 times faster than the sweeps performed by SLP (minimum sweep duration about 20 ms). The effects studied by those authors are not applicable to SLP/PICASSO.

Comment 4:
200 °C is a simulated temperature estimate for the metal rod itself, which is much higher than the temperature of the electronics font end (between 15°C and 45 °C along the orbit, measured).

Comment 5:
The simulation was done in flowing plasma (Vram = 7.5 km/s, perpendicular to the probe). This has been added to the manuscript.

The following text has also been added:
"The results presented in this paper qualitatively agree with those reported in (Ergun et al., 2021) and (Marholm et al., 2020), where it is shown that in the ionosphere, for cylindrical probes with finite length, the Beta exponent lies between the one of a perfect cylinder (0.5) and the one of a perfect sphere (1)."

"The fact that the main issue is the influence of the spacecraft potential rather than the wake due to the probe itself comes from the fact that the probe to S/C current collection area ratio is much larger for SLP on PICASSO than for LPW on MAVEN, which leads to more severe S/C charging effects for SLP. In addition, given the short distance between the S/C body and the sensor (a few Debye lengths) and the fact that there is no guard, the S/C potential effect dominates the wake effect on the I-V characteristics for SLP."

Comment 6:
The following text has been added:

"For a space instrument, one possibility would be to periodically stop sweeping, wait for a given time (to let the probe-spacecraft system discharge), restart the sweep sequence and use the first sweep as a reference. If the plasma environment can be assumed to be stable for a few tens of seconds, then the effects of contamination on the I-V curves can be accounted for in the calibration. Measuring with multiple probes would allow acquiring continuous measurements data even if sweeps are stopped and restarted regularly (if this is done alternatively for both - or more – probes).

Another possibility is to perform increasing and decreasing sweeps with different duration as in (Lebreton, et al., 2006) and to model the contamination layer as an R-C parallel circuit as in (Fang et al., 2018 and Piel et al., 2001). If properly modeled and combined with a database of PIC simulations of contaminated probe surfaces for different R, C and plasma parameters values, this could be used to retrieve the uncontaminated I-V curve. The study of the effect of the contamination, on a single sweep and on consecutive sweeps, is an on-going work that will be subject of a follow-up publication. Hence in this article, we acknowledge that there are uncertainties in the determination of the plasma parameters, but their quantification is out of scope of this paper."

Comment 7:
We added the following text:
"Similar effects (i.e. reduced current at a given probe potential, the derived plasma potential becomes more positive, the derived electron density decreases and the derived electron temperature becomes hotter) have been reported in (Samaniego, J. I. et al., 2018) for several probe coatings (including Gold and DAG) both when the probes were exposed to oxygen-rich environments and without exposure to oxygen-rich environment. In this latter case, the sources of contamination were assumed to be the deposition of vacuum pump oil and moisture from the air, similarly to the contamination of the SLP probes reported in this Section."

As suggested, the following reference has been added:
Samaniego, J. I., Wang, X., Andersson, L., Malaspina, D., Ergun, R. E., & Horányi, M. (2018). Investigation of coatings for Langmuir probes in an oxygen-rich space environment. Journal of Geophysical Research: Space Physics, 123, 6054– 6064. https://doi.org/10.1029/2018JA025563

**Referee 2:**

Comment 1:
We added the following text:
"The PICASSO mission ended in June 2022 because of issues with the platform. Although a limited amount of time has been allocated to the payload commissioning, it has been possible to validate all the measurement and diagnostic modes of SLP."

Comment 2:
We added the following text:

"The OML collection theory can be applied if, *inter alia*, the cylindrical probe radius is small compared to the Debye length and the length of the probe is much longer than the Debye length (in practice more than 10 times the Debye length). If all assumptions for the OML theory are met, the equations (1) and (2) below can be used to determine the electron density and temperature from the I-V characteristics. If those assumptions are not met, then the equations must be adapted. In principle, the electron retardation region obeys the OML theory, irrespective of the probe geometry, hence the electron temperature can always be retrieved, if one can properly isolate the electron current contribution to the I-V curve."

Comment 3:
Figure 1 has been replaced by a figure where the scale is kept constant throughout the bias range (no current multiplication).

Comment 4:
The text has been updated for coherence, it is PICASSO everywhere now.

Comment 5:
The assumptions for the simulations have been added in the text:
"The simulation model is a cube of aluminium (AL2K - Aluminium according to NASCAP-2k), each side 141.4 mm long, similarly to the model that has been tested in the plasma chamber, with a 40 mm long probe in aluminium (AL2K) located 60 mm away from the cube. The radius of the probe is 2 mm. The plasma density is set to 2e10 /m³ and the electron and ion temperature to 0.05 eV. The simulation is performed in flowing plasma (V = 7.5 km/s, perpendicular to the probe), without taking into account photoemission (worst-case in terms of S/C charging due to Langmuir probe). The absolute spacecraft capacitance is set to 200 pF, ions are treated as PIC and electrons are treated as a Maxwell Boltzmann equilibrium distribution."